# PLANNING WITH MCTS: ENHANCING PROBLEM-SOLVING IN LARGE LANGUAGE MODELS

## ABSTRACT

Despite recent advances in Large Language Models (LLMs), their ability to solve complex reasoning problems remains limited by inconsistent planning and logical flaws. We present a novel framework that significantly enhances LLMs' problem-solving capabilities by leveraging Monte Carlo Tree Search (MCTS) for plan generation. Unlike previous approaches that apply MCTS to solution search, our method uniquely integrates MCTS into the planning phase, guided by specialized LLM-powered agents that evaluate plan quality. Experiments across diverse benchmark datasets demonstrate that our approach improves problem-solving accuracy by an average of 40.59% compared to zero-shot Chain-of-Thought prompting. Furthermore, we show that using smaller models for MCTS planning and larger models for execution can maintain high performance while reducing computational costs. This work opens new avenues for developing more robust and efficient AI systems capable of tackling complex real-world problems, with potential applications in fields requiring advanced logical reasoning and long-term planning. Our code examples are publicly available at this Anonymous Github Repository.

## 1 INTRODUCTION

In an era where artificial intelligence increasingly influences our daily lives, the quest for machines capable of human-like reasoning remains a central challenge in computer science. Large Language Models (LLMs) have emerged as powerful tools in natural language processing, demonstrating remarkable capabilities across various tasks (Brown et al., 2020; Chowdhery et al., 2023). However, when faced with complex, multi-step problems requiring logical consistency and long-term planning, even state-of-the-art models like GPT-4 (OpenAI et al., 2024) often falter, revealing a critical gap between machine and human problem-solving abilities.

The limitations of LLMs in complex reasoning tasks stem from their struggle to maintain logical consistency and handle long-term dependencies throughout extended problem-solving processes. Existing approaches, such as Chain-of-Thought (CoT) prompting (Wei et al., 2022), have shown promise in improving step-by-step reasoning. However, these methods often fail to address the fundamental challenge of generating and following a coherent, overarching plan (Wang et al., 2023b; Yao et al., 2023). Similarly, current planning methods for LLMs, including task decomposition (Patel et al., 2022; Zhou et al., 2023) and explicit plan-and-solve approaches (Wang et al., 2023a; Yao et al., 2023), are ultimately constrained by the LLM's inherent reasoning abilities.

This research addresses a critical question: How can we enhance the planning capabilities of LLMs to significantly improve their problem-solving performance in complex, multi-step tasks? We hypothesize that by leveraging advanced search algorithms in the planning phase, we can generate higher-quality plans that guide LLMs towards more effective and logically consistent solutions.

To this end, we propose a novel framework that integrates Monte Carlo Tree Search (MCTS) into the planning process for LLMs. MCTS, renowned for its success in complex game-playing AI (Silver et al., 2016), offers a powerful method for exploring vast search spaces and identifying optimal strategies. Our approach uniquely applies MCTS to the generation of problem-solving plans, rather than to direct solution search. This is achieved through a two-step process: first, MCTS explores the space of possible plans, guided by specialized LLM-powered agents that evaluate plan quality; then, the optimal plan is provided to the LLM for step-by-step execution.

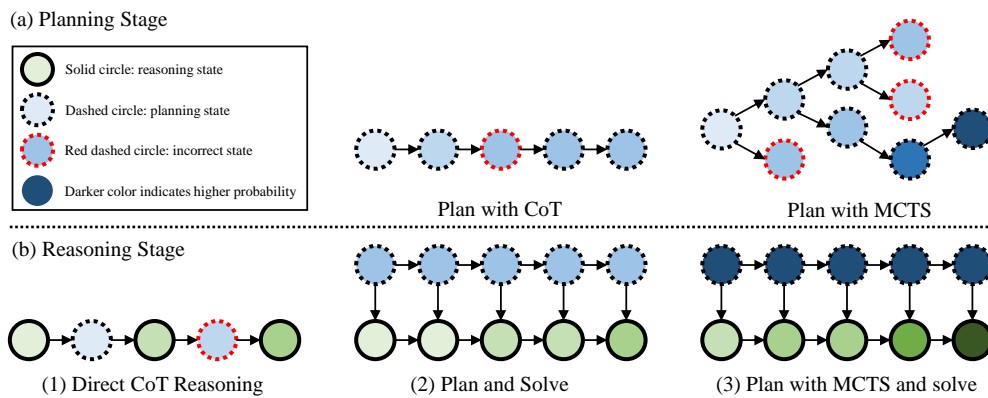

Figure 1: Illustration of the proposed framework for enhancing LLM problem-solving with MCTS planning. (a) CoT prompting generates planning and reasoning states interleavedly, while post states may suffer from cumulative error. (b) Plan and solve framework separates planning and reasoning, while planning is generated manually or based on CoT, which may not be optimal. (c) Our proposed MCTS-enhanced planning approach explicitly generates high-quality plans using MCTS, which are then executed by the LLM for problem-solving. Darker colors of each state indicate higher probability, which leads to better problem-solving performance.

The significance of this research extends beyond mere incremental improvements in LLM performance. By enhancing the planning and problem-solving capabilities of LLMs, we pave the way for more robust and reliable AI systems capable of tackling complex real-world problems. This has far-reaching implications for fields such as automated reasoning, decision support systems, and AI-assisted scientific discovery.

Our work makes several key contributions to the field of AI and natural language processing:

1. We introduce a novel paradigm for enhancing LLMs' problem-solving abilities by applying MCTS to the planning process, addressing a critical gap in current approaches.

2. We provide empirical evidence demonstrating significant improvements in problem-solving accuracy across diverse benchmark datasets, with an average improvement of 40.59% compared to zero-shot Chain-of-Thought prompting.

3. We offer insights into the interplay between planning, reasoning, and execution in LLM-based problem-solving, contributing to a deeper understanding of AI cognition.

4. We propose an efficient approach using smaller models for MCTS planning and larger models for execution, maintaining high performance while reducing computational costs.

## 2 Searching Plan for Better Problem-Solving

This section presents our novel approach to enhancing the problem-solving capabilities of LLMs through the integration of MCTS in the planning process. We first provide the necessary background and problem formulation, followed by a detailed description of our MCTS-enhanced planning framework.

### 2.1 Preliminary: From CoT to Planning

LLMs have demonstrated remarkable performance in various natural language processing tasks, yet their ability to solve complex, multi-step problems remains limited. Existing approaches, such as CoT prompting (Wei et al., 2022), often struggle to maintain logical consistency and handle long-term dependencies in reasoning chains, especially as problem complexity increases (Wang et al., 2023b; Yao et al., 2023).

Formally, we can view problem-solving with LLMs as a process of generating a solution $Y$ given a problem $X$ and a context $C$, which can be represented as a conditional probability distribution

---

**Count Characters: Example 1**

**Problem:** how many R's are there in the word "Strawberry"?
—
**CoT Prompt:** Let's think step by step.
**CoT Result:**
Here's how to count the 'R's in "strawberry":
1. **Identify the 'R's:** Look for the letter 'R' in the word.
2. **Count:** There are **two** 'R's in the word "strawberry".
—
**Planning Prompt:**
To count the R's in "strawberry." we can iterate throuoh each letter of the word, Initially, we'll set a counter to zero, for each leter encountered, we'll check if itis an "R" character. If it is, we'll increment our counter, This proess will continwe until we've examined all letters in the word, allowing us to determine the final count of "R's present.
**Planning Result:**
Following the plan:
1. **Initialize counter:** 'r_count = 0'.
2. **Iterate through the word:** Check each letter of "strawberry" for 'R'.
3. **Increment counter:** If the letter is 'R', increment the counter ('r_count = r_count + 1').
4. **Final Count:** The word "strawberry" contains **3** 'R's.
—
**MCTS Planning:**
1. **Define the task:** Our goal is to count the number of "R's present within the word "strawberry."
2. **Iterate through the word:** We will examine each letter of "strawberry" one at a time.
3. **Check for "R":** If the letter is an "R", increment a counter to keep track of our count.
4. **Increment counter:** If the letter is an "R", increment a counter to keep track of our count.
5. **Complete iteration:** We will continue this process until we have examined every letter.
6. **Return the count:** After processing all letters, the counter will hold the final number of "R's.
**MCTS Planning Result:**
Here's the breakdown:
1. **Define the task:** We need to count how many "R's are in "strawberry".
2. **Iterate through the word:** Let's examine each letter:
* **s** - Not an R, counter remains 0.
* **t** - Not an R, counter remains 0.
* **r** - **It's an R! Increment the counter to 1.**
* **a** - Not an R, counter remains 1.
* **w** - Not an R, counter remains 1.
* **b** - Not an R, counter remains 1.
* **e** - Not an R, counter remains 1.
* **r** - **It's an R! Increment the counter to 2.**
* **r** - **It's an R! Increment the counter to 3.**
* **y** - Not an R, counter remains 3.
3. **Complete iteration:** After examining all letters, the counter holds **3** R's.
4. **Result:** There are **3** R's in the word "strawberry".

---

Figure 2: Character counting example

$P(Y|X, C)$. In zero-shot CoT prompting, the context $C$ is typically the problem itself, and the LLM attempts to generate the solution directly through a series of reasoning steps $X = \{X_1, X_2, ..., X_n\}$. This direct approach often leads to suboptimal solutions due to the intertwined nature of planning and reasoning.

To address this challenge, we employ a planning-based approach that decouples planning from reasoning. We introduce a plan $\pi$ as an ordered sequence of actions, where $\pi \in \Pi$, and $\Pi$ represents the space of all possible plans. In this framework, the context $C$ can be decomposed into two parts: the problem description $C_{problem}$ and a plan $\pi$ generated based on the problem, which we denote as $C_{plan}$. Thus, we can rewrite the conditional probability as:

$$P(Y|X, C) = P(Y|X, C_{problem}, C_{plan}) \tag{1}$$

Assuming that the solution $Y$ depends on the problem $X$ and the plan $C_{plan}$, but not directly on the problem description $C_{problem}$ once the plan is given, we can factorize this probability using the chain rule:

$$P(Y|X, C_{plan}) = P(Y|X, C_{plan})P(X|C) \tag{2}$$

This factorization highlights the two distinct stages of our approach:

1. **Planning** ($P(X|C)$): Generating a sequence of reasoning steps (X) based on the initial context (C), which includes the problem description.

2. **Reasoning** ($P(Y|X, C_{plan})$): Generating the solution (Y) given the problem (X) and the plan ($C_{plan}$).

By explicitly separating these stages, we aim to improve the logical consistency and coherence of the problem-solving process. The planning stage focuses on finding an effective plan, while the reasoning stage leverages the plan to guide the generation of the solution.

Instead of relying on the LLM to implicitly generate a plan within its reasoning process, we leverage MCTS (Chaslot et al., 2008) to explicitly explore the plan space $\Pi$ during the planning stage. MCTS is a powerful heuristic search algorithm that has achieved remarkable success in complex game AI (Silver et al., 2016). We adapt MCTS to the problem-solving domain by representing plans as nodes in the search tree and using specialized LLM-powered agents to evaluate the quality of each plan. These agents assess properties such as the logical consistency and feasibility of the plan, providing feedback that guides the MCTS exploration.

This approach differs significantly from previous work that has applied MCTS to LLMs. While some studies have used Breadth First Search (BFS) or Depth First Search (DFS) to guide the selection of reasoning steps in CoT (Yao et al., 2023), our framework focuses on using MCTS to optimize the plan itself, providing a more structured and potentially more effective approach to complex problem-solving.

By decoupling planning and reasoning, and by leveraging the strengths of MCTS and LLMs, our proposed framework aims to enhance the problem-solving capabilities of LLMs, particularly in scenarios requiring complex reasoning and long-term planning.

### 2.2 MONTE CARLO TREE SEARCH FOR PLANNING

To address the limitations of existing planning methods and improve the coherence and logical consistency of LLM problem-solving, we propose leveraging MCTS for plan generation. Unlike approaches that use MCTS to directly guide the reasoning process, our approach focuses on finding a high-quality plan **before** the LLM begins to reason towards a solution. This corresponds to explicitly searching for a good sequence of reasoning steps ($X$) in the probabilistic framework $P(Y|X, C_{plan})$ discussed in Sec. 2.1.

**Selection:** Starting from the root node (which represents an initial plan, generated by prompting the LLM with the problem description), we traverse the tree by selecting the child node with the highest Upper Confidence Bound 1 (UCB1) value. UCB1 balances exploration and exploitation by considering both the average reward of simulations passing through a node and the number of times it has been visited:

$$UCB1(node) = Q(node) + C\sqrt{\frac{\ln(N(parent))}{N(node)}} \tag{3}$$

where $Q(node)$ is the average reward of simulations passing through the node, $N(node)$ is the number of times the node has been visited, $N(parent)$ is the number of times the parent node has been visited, and $C$ is an exploration constant that controls the balance between exploration and exploitation.

**Expansion:** When a leaf node is reached, a new node is added to the tree. This new node represents a modified version of the parent node's plan.

Plans are represented as sequences of natural language instructions that guide the LLM's reasoning. For example, a plan to solve a math word problem might be: "1. Identify the given quantities. 2. Determine the relationship between the quantities. 3. Formulate an equation. 4. Solve the equation."

**Simulation and Reward:** The newly generated plan is then subjected to a simulation to estimate its effectiveness. Instead of real execution in traditional MCTS (Chaslot et al., 2008; Silver et al., 2016), we employ multi-agents to assess the quality of the plan inspired by (Zhang et al., 2024b), including:

- **Logical Consistency Agent:** Checks for contradictions or inconsistencies in the plan.
- **Feasibility Agent:** Determines whether the plan is executable.

Each evaluation agent assigns a score (e.g., between 0 and 1) to the plan, and may also provide textual feedback explaining its assessment. For example, the Logical Consistency Agent might give a low score and feedback like: "Step 2 contradicts the information given in Step 1." This feedback can be used to guide plan modification in subsequent expansion steps.

The individual scores from the evaluation agents are combined by a reward function to produce an overall reward signal for the plan. The reward function can be a simple weighted average or a more complex function that takes into account the relative importance of different evaluation criteria.

**Backpropagation:** The reward signal obtained from the simulation and evaluation is backpropagated up the MCTS tree, updating the value estimates of all nodes along the path from the root to the newly expanded node.

By iteratively applying these steps, MCTS builds a search tree of potential plans, gradually focusing on areas of the plan space that are likely to lead to high-quality solutions. Once a predefined number of rollouts have been completed, or a time limit is reached, the MCTS algorithm selects the node with the highest average reward as the optimal plan. We present an illustrative example of the MCTS planning process in Fig. 2 with counting "R"s in the word "Strawberry" and a more complex example in Sec. A.1.

## 3 EXPERIMENTS

### 3.1 RESEARCH QUESTIONS

In this research, we would like to investigatee the potential of MCTS to enhance the planning and problem-solving capabilities of LLMs, addressing limitations of existing methods like zero-shot CoT (Kojima et al., 2022) and Plan-and-Solve (Wang et al., 2023a) prompting in handling complex multi-step problems. Specifically, we explore two core research questions:

**RQ1:** *Does MCTS-based planning improve LLM problem-solving performance?* We will evaluate this by comparing the accuracy of our proposed approach against baseline prompting and analyzing the correlation between plan quality and LLM performance.

**RQ2:** *How to optimize the MCTS parameters for planning in LLMs?* We will investigate the impact of different MCTS strategy, such as different LLM for planning, evaluating, and executing, on the problem-solving performance of our approach.

By answering these questions, we aim to provide insights into the interplay between planning and reasoning in LLMs, contributing to the development of more robust and reliable AI systems capable of tackling complex real-world problems.

### 3.2 DATASET AND EVALUATION METRICS

**Benchmark Datasets:** To evaluate the effectiveness of our MCTS-enhanced planning approach, we conduct experiments on a diverse set of benchmark datasets specifically chosen to assess its performance across various problem-solving tasks and challenge the limitations of existing methods, including (1) arithmatic: GSM8K (Cobbe et al., 2021), AddSub (Hosseini et al., 2014), Multi-Arith (Roy & Roth, 2015), SVAMP (Patel et al., 2021), and SingleEq (Koncel-Kedziorski et al., 2015) (2) commonsense reasonin: CommonsensQA (Talmor et al., 2019) (3) symbolic reasoning: Last Letters (Wei et al., 2022) (4) Gaming reasoning: Object Tracking (Srivastava et al., 2023).

With these datasets, we aim to cover a wide range of problem-solving tasks, including arithmetic, commonsense reasoning, symbolic reasoning, and gaming reasoning, to evaluate the generalizability of our approach.

**Evaluation Metrics:** We employ accuracy as our primary evaluation metric across all datasets, measuring the percentage of correctly solved problems. Accuracy provides a direct measure of the LLM's problem-solving capabilities and allows for straightforward comparison between different approaches.

Table 1: Comparison of MCTS Planning and Zero-shot CoT on different datasets.

| Type | Dataset
Model | AddSub | CommonsensQA | GSM8K | Last Letters | MultiArith | Object Tracking | SingleEq | SVAMP |
|---|---|---|---|---|---|---|---|---|---|
| Zero-shot CoT | Qwen2.5-7B-Instruct | 85.06 | 63.72 | 80.89 | 21.00 | 95.33 | 74.80 | 77.17 | 83.40 |
| ( (Kojima et al., 2022)) | Meta-Llama-3.1-8B-Instruct | 28.61 | 63.80 | 57.32 | 26.40 | 38.17 | 49.33 | 39.76 | 27.00 |
| CoT Plan | Qwen2.5-0.5B-Instruct | 36.96 | 31.20 | 17.82 | 0.00 | 37.00 | 31.87 | 44.09 | 29.90 |
| ( (Wang et al., 2023a)) | Qwen2.5-1.5B-Instruct | 28.61 | 44.23 | 35.03 | 7.20 | 44.67 | 29.87 | 34.84 | 33.90 |
| | Qwen2.5-7B-Instruct | 87.59 | 78.62 | 88.84 | 55.20 | 98.33 | 79.33 | 93.70 | 91.90 |
| | Meta-Llama-3.1-8B-Instruct | 78.23 | 57.14 | 74.77 | 15.40 | 91.58 | 57.94 | 84.65 | 79.20 |
| MCTS Plan (Ours) | Qwen2.5-0.5B-Instruct | 58.23 | 32.43 | 29.80 | 0.20 | 68.33 | 27.20 | 69.69 | 43.80 |
| | Qwen2.5-1.5B-Instruct | 75.70 | 58.72 | 64.29 | 6.40 | 86.50 | 26.40 | 83.66 | 72.10 |
| | Qwen2.5-7B-Instruct | 88.10 | 79.20 | 90.14 | 56.60 | 98.67 | 79.33 | 92.91 | 92.90 |
| | Meta-Llama-3.1-8B-Instruct | 80.51 | 68.57 | 77.28 | 12.80 | 92.76 | 55.43 | 87.99 | 81.20 |
| CoT Plan Avg. | | 57.85 | 52.80 | 54.12 | 19.45 | 67.89 | 49.75 | 64.32 | 58.73 |
| MCTS Plan Avg. | | 75.63 | 59.73 | 65.38 | 19.00 | 86.56 | 47.09 | 83.56 | 72.50 |
| Changes | | +17.79 | +6.93 | +11.26 | -0.45 | +18.67 | -2.66 | +19.24 | +13.77 |

## 3.3 RQ1: Enhanced Problem-Solving through MCTS-Guided Planning

Existing methods for enhancing LLM problem-solving often face challenges in maintaining logical consistency and handling long-term dependencies in complex multi-step problems. This is primarily because these methods rely heavily on the LLM's inherent reasoning capabilities, which can be limited in such scenarios. We hypothesize that applying MCTS to the planning process can address these limitations by generating higher-quality, more logically sound plans that guide the LLM towards more effective solutions. MCTS excels at exploring large search spaces and identifying optimal strategies through its balance of exploration and exploitation. By leveraging MCTS to generate plans, we aim to overcome the inherent limitations of relying solely on the LLM's reasoning for planning.

To investigate the impact of MCTS-generated plans on LLM problem-solving, we conduct experiments comparing the performance of our MCTS-enhanced planning approach against two baselines: (1) standard CoT prompting (Kojima et al., 2022) and (2) a plan-and-solve approach (Wang et al., 2023a) where the LLM first generates a plan and then executes it. We evaluate these methods on a diverse set of datasets introduced in Sec. 3.2.

We employ two state-of-the-art open-sourced LLMs: LLama 3.1 (8B parameters) (Dubey et al., 2024) and Qwen 2.5 (0.5B, 1.5B, and 7B parameters) (Yang et al., 2024). These models represent a strong baseline for current LLM capabilities and allow for a fair comparison between different planning approaches. Notably, we include smaller variants of Qwen 2.5 (0.5B and 1.5B) for MCTS to investigate the impact of model size on planning performance and computational efficiency, especially considering the potentially high computational cost of MCTS.

We adopt a zero-shot setting for both CoT and MCTS to assess the methods' ability to generalize to new problems without task-specific fine-tuning. For the CoT baseline, we use code and data from (Kong et al., 2024) and (Kojima et al., 2022). We leverage the SGLang platform (Zheng et al., 2024) for hosting and interacting with the LLMs.

Tab. 1 presents the comparison of MCTS Planning and Zero-shot CoT on different datasets. Overall, the MCTS-enhanced planning approach outperforms the CoT baseline across most datasets, with an average improvement of 40.59%. This substantial improvement highlights the effectiveness of MCTS in generating high-quality plans for LLMs. As expected, larger models generally perform better across all methods. However, the performance gap between small and large models is narrower for MCTS Planning compared to CoT, suggesting that our approach can partially compensate for the limitations of smaller models.

The improvements are particularly pronounced in tasks that require complex reasoning and long-term planning. For example, on arithmetic tasks (MultiArith, SingleEq, and AddSub), MCTS Planning shows average improvements of 18.67%, 19.24%, and 17.79%, respectively. Similarly, on problem-solving tasks (GSM8K and SVAMP), we observe improvements of 11.26% and 13.77%, respectively.

These results strongly support our hypothesis that MCTS can enhance LLM problem-solving by providing more structured and coherent plans that guide the reasoning process effectively. The

Table 2: Performance Comparison of Language Models

| Model | Max Depth | | | | | | | | Number of Rollouts | | | | | |
| --- | --- | --- | --- | --- | --- | --- | --- | --- | --- | --- | --- | --- | --- | --- |
| | 1 | 3 | 5 | 7 | 10 | 20 | 50 | 100 | 1 | 3 | 5 | 7 | 10 | 20 |
| Meta-Llama-3.1-8B-Instruct | 74.22 | 75.97 | 76.12 | 76.42 | 76.80 | 76.42 | 77.18 | 77.41 | 79.08 | 76.88 | 76.04 | 76.57 | 76.19 | 77.48 |
| Qwen2.5-7B-Instruct | 87.64 | 88.17 | 88.02 | 88.40 | 88.32 | 88.48 | 88.55 | 88.78 | 89.01 | 89.76 | 89.08 | 89.23 | 90.14 | 89.92 |

Table 3: Performance Cmparison of Different Evaluation Agents for MCTS.

| Model | Evaluator | AddSub | CommonsensQA | GSM8K | Last Letters | MultiArith | Object Tracking | SingleEq | SVAMP |
| --- | --- | --- | --- | --- | --- | --- | --- | --- | --- |
| Qwen2.5-7B-Instruct | Feasibility | 88.1 | 71.3 | 89.5 | 58.4 | 97.7 | 65.5 | 91.5 | 92.2 |
| Qwen2.5-7B-Instruct | Logical Consistency | 86.6 | 70.9 | 89.2 | 58.2 | 97.2 | 65.0 | 91.5 | 91.4 |
| Qwen2.5-7B-Instruct | Combined (Ours) | 88.1 | 79.2 | 90.1 | 56.6 | 98.7 | 79.3 | 92.9 | 92.9 |

superior performance in complex tasks suggests that MCTS is particularly adept at decomposing multi-step problems and maintaining logical consistency throughout the solution process.

To investigate the key factors influencing the effectiveness of MCTS in planning for LLMs, we conduct an ablation study to analyze the impact of different components of our approach. Specifically, we evaluate the following factors on the GSM8K dataset: (1) the maximum depth of the search tree, (2) the number of rollouts in MCTS, and (3) different evaluation agents for reward computation.

Tab. 2 illustrates the results of our ablation study. We observe that:

**Depth:** The performance of MCTS Planning generally improves with increasing depth, indicating that deeper search trees allow for more thorough exploration of the plan space and lead to higher-quality plans. The rate of improvement slows down as depth increases, suggesting that the plan can be easily optimized in the first few steps, while later steps are more difficult to optimize. This behavior implies that the MCTS may early-stop the search when evaluator agents give high scores to the plan.

**Rollouts:** The number of rollouts also impacts the performance of MCTS Planning. Initially, increasing the number of rollouts from a small number yields significant improvements. However, the uncertainty of performance improvement increases as the number of rollouts grows, indicating that with an expanding search space and higher computational cost, the performance gains may diminish. We implemented backpropagate in a zero-sum game manner, which may contribute to the increased uncertainty in performance improvement. Future work could explore alternative improvement strategies to address this issue.

**Evaluation Agents:** We tested different evaluation agents for computing rewards in the MCTS process. Tab. 3 shows the results of using feasibility and logical consistency evaluators compared to our final approach. While both evaluators improved performance over the baseline, our combined approach yielded the best results across most datasets.

These results demonstrate the effectiveness of our MCTS-guided planning approach in enhancing the problem-solving capabilities of LLMs. By generating high-quality plans through MCTS, we enable LLMs to tackle complex reasoning tasks more effectively, maintaining logical consistency and coherence throughout the problem-solving process. However, it is important to note that the computational cost of MCTS may be a limiting factor in some applications, and future work should explore ways to optimize the algorithm for efficiency without sacrificing performance.

### 3.4 RQ2: Enhancing the Efficiency of MCTS in Planning for LLMs

As demonstrated in Sec. 3.3, MCTS-based planning can significantly improve the problem-solving performance of LLMs. However, the computational cost of MCTS poses challenges for real-world applications, particularly when using large models for complex problem-solving tasks. Interestingly, our observations from Tab. 1 suggest that smaller models can benefit substantially from MCTS planning. This finding presents a promising direction for optimizing the efficiency of MCTS in planning for LLMs. We therefore pose the following research question: **Can we leverage small LLMs for optimal plan search and large LLMs for plan execution to enhance the efficiency and effectiveness of MCTS-based planning for LLMs?**

Table 4: Performance Comparison of Different LLMs for Planning and Execution.

| Planner Model | Evaluator Model | Dataset Executor Model | AddSub | CommonsensQA | GSM8K | Last Letters | MultiArith | Object Tracking | SingleEq | SVAMP |
|---|---|---|---|---|---|---|---|---|---|---|
| Qwen2.5-1.5B-Instruct | Qwen2.5-1.5B-Instruct | Qwen2.5-1.5B-Instruct | 75.70 | 58.72 | 64.29 | 6.40 | 86.50 | 26.40 | 83.66 | 72.10 |
| gemma-2-2b-it | gemma-2-2b-it | gemma-2-2b-it | 81.52 | 17.16 | 52.67 | 4.47 | 86.83 | 20.00 | 86.02 | 69.40 |
| Qwen2.5-1.5B-Instruct | Qwen2.5-1.5B-Instruct | Qwen2.5-72B-Instruct | 88.86 | 78.49 | 86.96 | 47.20 | 96.83 | 80.40 | 95.28 | 90.70 |
| | Qwen2.5-72B-Instruct | Qwen2.5-1.5B-Instruct | 86.58 | 68.36 | 81.35 | 28.20 | 92.50 | 47.12 | 88.78 | 81.40 |
| | Qwen2.5-72B-Instruct | Qwen2.5-72B-Instruct | 90.63 | 80.71 | 92.80 | 76.80 | 98.67 | 89.47 | 94.88 | 92.00 |
| gemma-2-2b-it | gemma-2-2b-it | Qwen2.5-72B-Instruct | 91.39 | 77.72 | 88.48 | 69.40 | 97.67 | 83.87 | 95.67 | 92.30 |
| | Qwen2.5-72B-Instruct | gemma-2-2b-it | 90.89 | 34.64 | 79.83 | 54.60 | 95.00 | 35.87 | 92.13 | 84.00 |
| | Qwen2.5-72B-Instruct | Qwen2.5-72B-Instruct | 92.41 | 78.54 | 92.42 | 78.20 | 98.33 | 80.53 | 95.87 | 93.00 |
| Qwen2.5-72B-Instruct | Qwen2.5-72B-Instruct | Qwen2.5-72B-Instruct | 91.14 | 83.95 | 94.62 | 85.60 | 98.67 | 97.86 | 95.08 | 93.40 |

To investigate this question, we designed an experimental setup using two small LLMs (Qwen2.5-1.5B-Instruct and Gemma-2-2b-it (Team et al., 2024)) for plan search, and a large LLM (Qwen2.5-72B-Instruct) for plan evaluation or execution. We evaluated this approach across the benchmark datasets introduced in Sec. 3.2, comparing various combinations of small and large models for planning, evaluation, and execution tasks.

Tab. 4 presents a comprehensive performance comparison of different LLM combinations for planning and execution tasks. Our key findings include:

**Small-Large Model Synergy:** Using a small model for planning and a large model for execution significantly improved problem-solving performance across all datasets. For instance, the Qwen2.5-1.5B-Instruct (planner) + Qwen2.5-72B-Instruct (evaluator+executor) combination achieved an average improvement of 23.87% compared to using Qwen2.5-1.5B-Instruct alone.

**Efficiency Gains:** The small-large model combination approached the performance of the large model (Qwen2.5-72B-Instruct) used alone, while potentially offering significant computational savings during the planning phase. For example, on the GSM8K dataset, the small-large combination achieved 92.80% accuracy, compared to 94.62% for the large model alone.

**Model-Specific Performance:** Interestingly, the Gemma-2-2b-it model, despite its smaller size, showed competitive performance when used for planning. This suggests that model architecture and training, not just size, play crucial roles in planning effectiveness.

Our findings demonstrate that leveraging small LLMs for planning and large LLMs for execution can significantly enhance the efficiency of MCTS-based planning while maintaining high problem-solving performance. This approach offers several advantages:

1. **Computational Efficiency:** By using smaller models for the computationally intensive planning phase, we can reduce the overall computational requirements without significantly sacrificing performance.

2. **Scalability:** This method allows for more efficient scaling of MCTS-based planning to larger and more complex problem spaces.

3. **Resource Optimization:** Organizations can optimize their use of computational resources, potentially reducing costs and environmental impact.

However, it's important to note potential limitations, such as the need for careful model selection and the possibility of suboptimal plans due to the use of smaller models in the planning phase. Future work should investigate these trade-offs more thoroughly and explore techniques to mitigate potential drawbacks.

In conclusion, our research demonstrates a promising approach to enhancing the efficiency and effectiveness of MCTS-based planning for LLMs, opening new avenues for improving AI problem-solving capabilities in resource-constrained environments.

# 4 RELATED WORK

## 4.1 CHAIN-OF-THOUGHT PROMPTING

Chain-of-Thought (CoT) prompting has emerged as a prominent technique to improve the reasoning abilities of LLMs by encouraging step-by-step problem-solving (Wei et al., 2022). By decomposing complex tasks into smaller, manageable steps, CoT aims to enhance logical consistency and reduce errors in multi-step reasoning. Subsequent studies have explored various enhancements to CoT, such as self-consistency approaches (Wang et al., 2023b) and tree-of-thought methods (Yao et al., 2023), which attempt to maintain logical coherence over extended reasoning paths. Despite these advancements, CoT methods often struggle with maintaining overall logical flow in highly complex scenarios, leading to inconsistencies and suboptimal solutions. How to maintain logical consistency and coherence in multi-step reasoning tasks is a research frontier in the field.

## 4.2 PLANNING TECHNIQUES FOR LLMS

Beyond CoT, several planning-based approaches have been proposed to bolster the problem-solving capabilities of LLMs. Task decomposition techniques (Patel et al., 2022; Zhou et al., 2023) involve breaking down complex problems into simpler sub-tasks, which the LLM can solve sequentially. Explicit plan-and-solve frameworks (Wang et al., 2023a; Besta et al., 2024) require the LLM to generate a plan before executing it, aiming to structure the reasoning process more effectively. Advanced methods like Skeleton-of-Thoughts (Ning et al., 2024) and Graph-of-Thought (Besta et al., 2024) introduce more sophisticated representations of plans to capture dependencies and improve coherence. However, these methods remain constrained by the LLM's inherent reasoning limitations, often failing to produce optimal plans in the face of complex, multi-step problems. How to generate high-quality plans for LLMs remains a challenging research question in the field.

## 4.3 MONTE CARLO TREE SEARCH

MCTS is a heuristic search algorithm renowned for its success in game-playing AI, particularly in games with vast search spaces like Go and Chess (Silver et al., 2016). MCTS operates through four main phases: selection, expansion, simulation, and backpropagation (Chaslot et al., 2008). Its ability to balance exploration and exploitation makes it highly effective in navigating large decision trees to identify optimal strategies. Recent research has begun to explore the application of MCTS in the context of LLMs, particularly for solution search in CoT processes (OpenAI, 2024) or self-training process (Zhang et al., 2024a). These studies have demonstrated that MCTS can enhance the search for high-quality solutions by efficiently exploring the space of possible reasoning paths. However, the direct application of MCTS to LLMs presents challenges, such as the computational expense of evaluating generated content and the difficulty in effectively integrating heuristic evaluations within the planning process. We investigate these challenges in our research and propose to use smaller LLM for planning and larger LLM for evaluation and execution to address these challenges.

## 4.4 RESEARCH GAP AND OUR CONTRIBUTION

While MCTS has shown promise in enhancing solution search for LLMs, its application to the planning process remains largely unexplored. Existing planning methods for LLMs, including Chain-of-Thought prompting and plan-and-solve frameworks, predominantly rely on the LLM's inherent reasoning abilities to generate plans. However, these reasoning abilities can be inconsistent and prone to errors, particularly when dealing with complex, multi-step problems requiring long-term planning and logical consistency. For instance, in tasks involving multi-step mathematical reasoning or intricate commonsense scenarios, LLMs often struggle to devise and maintain a coherent plan, leading to suboptimal or incorrect solutions.

Our research addresses this gap by integrating MCTS into the planning phase. Unlike previous work focusing on solution search, we leverage MCTS to explicitly generate high-quality plans before the LLM starts reasoning. By systematically exploring the plan space and evaluating quality with specialized LLM agents, we aim to generate more effective and logically consistent plans, significantly enhancing LLM problem-solving accuracy and reliability in complex tasks.

## 5 CONCLUSION AND DISCUSSION

This paper introduced a novel framework for enhancing the problem-solving capabilities of LLMs by leveraging MCTS for plan generation. Our approach explicitly separates the planning and reasoning stages of problem-solving, using MCTS to explore the space of possible plans before the LLM begins to reason towards a solution. Experiments on a diverse set of benchmark datasets demonstrated that MCTS-enhanced planning significantly improves LLM problem-solving accuracy compared to standard Chain-of-Thought prompting and existing plan-and-solve methods.

Our findings highlight the effectiveness of MCTS in generating high-quality, logically consistent plans that guide the LLM towards more effective solutions, particularly in complex reasoning tasks. We also observed that MCTS planning offers a stronger performance boost for smaller LLMs, suggesting it can partially compensate for limited reasoning capabilities in resource-constrained settings. Further analysis revealed that the performance of MCTS planning is influenced by factors such as search depth and the number of rollouts, with deeper search and more rollouts generally leading to better plans.

To address the computational cost of MCTS, we explored the use of smaller LLMs for planning and larger LLMs for evaluation and execution. Our results demonstrated that this approach can significantly enhance efficiency while maintaining high problem-solving performance, offering a promising direction for optimizing MCTS planning in real-world applications.

We believe that our research contributes to a deeper understanding of the interplay between planning and reasoning in LLMs and opens new avenues for improving AI problem-solving capabilities. The integration of MCTS and LLMs holds significant promise for developing more robust and reliable AI systems capable of tackling complex real-world problems.

## ETHICAL CONSIDERATIONS

This research enhances LLM problem-solving via MCTS planning. While our benchmark datasets pose minimal direct ethical concerns, the potential impact of improved AI problem-solving necessitates broader ethical consideration.

Integrating MCTS introduces unique challenges. Unlike CoT methods, MCTS explores a wider range of plans, some potentially ethical yet logically sound. For instance, directly plan and executing a toxic action could be rejected by LLM with alignment to ethical principles. However, if the search objective is to find such a plan, MCTS may escape the LLM's ethical constraints. In this context, Mitigation strategies warrant investigation. Incorporating fairness constraints into MCTS, adversarial training for agents, and human-in-the-loop plan review could enhance ethical outcomes.

## LIMITATIONS

While our MCTS-enhanced planning approach demonstrates significant improvements in LLM problem-solving capabilities, it is important to acknowledge several limitations of our current work.

Firstly, the computational cost of MCTS, particularly for larger language models, remains a significant challenge. Although we have shown that using smaller models for planning can mitigate this issue to some extent, further research is needed to optimize the efficiency of MCTS in the context of LLMs. Future work could explore pruning techniques or more sophisticated heuristics to reduce the search space without compromising plan quality.

Secondly, our study primarily focused on a specific set of benchmark datasets. While these datasets cover a range of problem types, they may not fully represent the diversity of real-world problems that LLMs might encounter. Expanding our evaluation to a broader set of tasks and domains would provide a more comprehensive assessment of our method's generalizability and robustness.

Despite these limitations, we believe that our work represents a significant step forward in enhancing the problem-solving capabilities of LLMs. By explicitly addressing these challenges, we hope to inspire further research that will lead to even more powerful and reliable AI systems capable of tackling complex real-world problems.

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

# A APPENDIX

## A.1 COUNT CHARACTERS: EXAMPLE

---

**Count Characters: Example 2.1**

**Problem:** how many R's are there in the word "Pneumonoultramicroscopicsilicovolcanoconiosis"?
—
**CoT Prompt:** Let's think step by step.
**CoT Result:**
1. **Identify the letter:** We're looking for the letter 'R'.
2. **Scan the word:** Carefully read through the word "Pneumonoultramicroscopicsilicovolcanoconiosis".
3. **Count the occurrences:** Count each instance of the letter 'R'. There is only one 'R'.
Therefore, there is only **one** 'R' in the word "Pneumonoultramicroscopicsilicovolcanoconiosis".
— **Planning Prompt:**
To count the R's in "Pneumonoultramicroscopicsilicovolcanoconiosis." we can iterate throuoh each letter of the word, Initially, we'll set a counter to zero, for each leter encountered, we'll check if itis an "R" character. If it is, we'll increment our counter, This proess will continwe until we've examined all letters in the word, allowing us to determine the final count of "R's present.
**Planning Result:**
Following the plan:
1. **Initialization:** Set a counter to zero (e.g., 'r_count = 0').
2. **Iteration:** Iterate through each letter of the word "Pneumonoultramicroscopicsilicovolcanoconiosis".
3. **Comparison:** For each letter, check if it is equal to "R".
4. **Increment:** If the letter is "R", increment the counter ('r_count = r_count + 1').
5. **Final Count:** After iterating through all letters, the value of 'r_count' will be the total number of "R"s.
Let's apply this to the word:
The word "Pneumonoultramicroscopicsilicovolcanoconiosis" contains **one** "R".

---

Figure 3: Count Characters Example of "R" in the word "Pneumonoultramicroscopicsilicovolcanoconiosis" with CoT and Planning.

**Count Characters: Example 2.2**

**MCTS Planning Prompt:**
1. **Define the task:** Our goal is to count the number of "R's present within the word "strawberry."
2. **Iterate through the word:** We will examine each letter of "strawberry" one at a time.
3. **Check for "R":** If the letter is an "R", increment a counter to keep track of our count.
4. **Increment counter:** If the letter is an "R", increment a counter to keep track of our count.
5. **Complete iteration:** We will continue this process until we have examined every letter in the word.
6. **Return the count:** After processing all letters, the counter will hold the final number of "R's in the word.

**MCTS Planning Result:**
Okay, let's adapt your plan to count the R's in "Pneumonoultramicroscopicsilicovolcanoconiosis":

**Define the task:** Our goal is to count the number of "R's" present within the word "Pneumonoultramicroscopicsilicovolcanoconiosis".

**Iterate through the word:** We will examine each letter of "Pneumonoultramicroscopicsilicovolcanoconiosis" one at a time.

Let's start with a counter set to 0.
* **P** - Not an R, counter remains 0.
* **n** - Not an R, counter remains 0.
* **e** - Not an R, counter remains 0.
* **u** - Not an R, counter remains 0.
* **m** - Not an R, counter remains 0.
* **o** - Not an R, counter remains 0.
* **n** - Not an R, counter remains 0.
* **o** - Not an R, counter remains 0.
* **u** - Not an R, counter remains 0.
* **l** - Not an R, counter remains 0.
* **t** - Not an R, counter remains 0.
* **r** - **It's an R! Increment the counter to 1.**
* **a** - Not an R, counter remains 1.
* **m** - Not an R, counter remains 1.
* **i** - Not an R, counter remains 1.
* **c** - Not an R, counter remains 1.
* **r** - **It's an R! Increment the counter to 2.**
* **o** - Not an R, counter remains 2.
* **s** - Not an R, counter remains 2.
* **i** - Not an R, counter remains 2.
* **l** - Not an R, counter remains 2.
* **i** - Not an R, counter remains 2.
* **c** - Not an R, counter remains 2.
* **o** - Not an R, counter remains 2.
* **v** - Not an R, counter remains 2.
* **o** - Not an R, counter remains 2.
SKIP FOR SPACE

**Complete iteration:** We have now processed all letters, and the counter holds the final number of "R's" in the word.
**Result:** There are **2** R's in the word "Pneumonoultramicroscopicsilicovolcanoconiosis".

Figure 4: Count Characters Example of "R" in the word "Pneumonoultramicroscopicsilicovolcanoconiosis" with MCTS Planning.

