# OpenReview forum: "Planning with MCTS: Enhancing Problem-Solving in Large Language Models"
_ICLR.cc/2025/Conference — ICLR 2025 Conference Withdrawn Submission_

### Official Review · Reviewer_go53 · 2024-10-31

**Soundness:** 2
**Presentation:** 3
**Contribution:** 3
**Rating:** 3
**Confidence:** 4

**Summary:**

This paper proposes to use Monte Carlo Tree Search (MCTS) in the planning stage of using LLM to solve reasoning problems. The authors use comprehensive experiments to show that the proposed method can be better than CoT prompting. The authors also discuss the possibility of using smaller model to plan while using a larger model to evaluate and execute.

**Strengths:**

- The idea of using MCTS in the planning stage is relatively novel compared to many recent works on directly using MCTS in the reasoning stage.
- The proposed algorithm is tested on a diverse combination of models and datasets.

**Weaknesses:**

- While the cost is mentioned multiple times in the paper, there are no results showing the cost. This leads to two major problems: 1). how much motivation do we have to use the smaller LLM to generate a plan and let the bigger model execute? How much cost is it saving from the original setting, and how much cost is it requiring compared to CoT prompting? 2). If the cost is relatively high, can MCTS perform better than simply sampling N times and choosing the best one, i.e, BoN based on reward?
- The comparison in the experiment does not seem to be strong enough to me. Given that MCTS is already spending more cost, other popular methods used in reasoning that could improve the performance with some cost should also be considered, for example, ReAct [1], Reflexion [2]. It is also noteworthy that there is some work on prompt engineers that use MCTS or "gradient descent" [3, 4, 5]. While the domain does not seem directly applicable, a comparison is still needed, or clear clarification should be included in the paper.
- While the paper is overall easy to follow, the reproducibility of the paper is unclear as there are many important details missing, including the prompt used for many important implementation designs. Please refer to my questions for some examples. The discussion on certain failing cases is also highly recommended, especially since this is not even covered in the limitation section of the paper.

[1]. Yao S, Zhao J, Yu D, et al. React: Synergizing reasoning and acting in language models[J]. arXiv preprint arXiv:2210.03629, 2022.

[2]. Shinn N, Cassano F, Gopinath A, et al. Reflexion: Language agents with verbal reinforcement learning[J]. Advances in Neural Information Processing Systems, 2024, 36.

[3]. Wang X, Li C, Wang Z, et al. PromptAgent: Strategic Planning with Language Models Enables Expert-level Prompt Optimization[C]//The Twelfth International Conference on Learning Representations.

[4]. Chen W, Koenig S, Dilkina B. RePrompt: Planning by Automatic Prompt Engineering for Large Language Models Agents[J]. arXiv preprint arXiv:2406.11132, 2024.

[5]. Pryzant R, Iter D, Li J, et al. Automatic Prompt Optimization with “Gradient Descent” and Beam Search[C]//Proceedings of the 2023 Conference on Empirical Methods in Natural Language Processing. 2023: 7957-7968.

**Questions:**

1. While an ablation study is included in Tab. 2 in the paper, what is the default parameter used for the experiments in Tab. 1 and Tab. 3?
2. How is Q initialized? Is it all 0?
3. How is the expansion done? By another LLM? The prompt used is needed in this case.
4. While I believe that the cost of the proposed method is not so cheap, I am also interested to see what the performance is compared to directly using LLM in the reasoning stage, as given the cost constraint, these two directions probably cannot be used in the meantime.
5. Why is object tracking such a failure for the proposed method? It seems that in all models, MCTS is no better than CoT.

---

> ### Author Response · Authors · 2024-11-28
> **Response to Q1 and Q2**
>
> ## Q1: Default Parameters in Experiments
>
> Thank you for pointing out the need for clarity regarding default parameters. The default parameters used in Tables 1 and 3 are as follows:
> - **Exploration Weight:** 1.0.
> - **Maximum Depth:** 10.
> - **Number of Rollouts:** 8.
>
> These values were chosen based on preliminary experiments to balance computational cost and plan quality. We acknowledge the importance of documenting these details and will include them in the revised manuscript under the Experimental Setup section.
>
> ## Q2: Initialization of $ Q $
>
> The $ Q $ values in MCTS are initialized to zero for all nodes. This choice aligns with standard MCTS practices, ensuring unbiased initial exploration. We agree that this detail should be explicitly mentioned and will add it to the MCTS methodology description in Sec.2.3 of the revised manuscript.

---

> ### Author Response · Authors · 2024-11-28
> **Response to Q3: Expansion Phase in MCTS-based Framework**
>
> Thank you for your thoughtful question regarding the expansion phase of our MCTS-based framework. Below, we provide a detailed explanation of how the expansion is conducted and clarify the role of LLMs and prompts in this process:
>
> 1. Expansion Process in MCTS:
>     During the expansion phase of MCTS, a new node (representing a modified plan) is added to the search tree. The modification is guided by feedback collected during previous rollouts. The steps are as follows:
>     - A leaf node is selected based on the Upper Confidence Bound (UCB1) criterion.
>     - The existing plan at the leaf node is modified to address areas identified as suboptimal or inconsistent, such as logical contradictions or missing steps.
>     - The modified plan becomes the new child node in the tree.
>
> 2. Role of LLMs in Expansion:
>     The LLM plays a critical role in generating and refining plans:
>     - **Initial Plan Generation:** The root node of the tree is initialized with a plan generated by prompting the LLM to create a baseline solution based on the problem description.
>     - **Plan Modification (Expansion):** The LLM is prompted to revise the existing plan using specific feedback from evaluation agents (e.g., Logical Consistency Agent). The prompt structure includes:
>       - The original plan.
>       - Feedback highlighting issues or suggesting improvements.
>       - A request to modify the plan while addressing the feedback.
>     - This iterative process allows the LLM to refine plans step-by-step, ensuring logical consistency and feasibility.
>
> 3. Prompts example in Expansion:
>     Below is an example prompt used during the expansion phase:
>     ```
>     # Problem
>     [Problem Description]
>
>     # Current Plan
>     [Existing Plan]
>
>     # Feedback
>     - Logical Consistency: [Feedback on contradictions or inconsistencies]
>     - Feasibility: [Feedback on execution feasibility]
>
>     # Task
>     Modify the current plan to address the feedback provided. Ensure that the revised plan is:
>     - Logically consistent.
>     - Feasible for execution.
>     - Concise and actionable.
>
>     Provide the modified plan in the following format:
>     ---
>     # Revised Plan
>     [Modified Plan]
>     ---
>     ```
>     This ensures transparency and reproducibility of the plan modification process. You can find the detailed prompt implementation in our [Anonymous GitHub Repository](https://github.com/Anonymous-gwFabfaH/MCTS-Planning/blob/main/src/MCTS_planning.py#L99-L111)
>
> 4. Commitment to Address Prompt Details:
>     We acknowledge that the initial manuscript did not provide sufficient details about the prompts used in the expansion phase. We will include:
>     - Specific examples of prompts used for initial plan generation and expansion.
>     - Details on how feedback from evaluation agents informs prompt design.
>     These additions will enhance the transparency and reproducibility of our methodology.
>
> By clarifying these points, we hope to address your concern comprehensively. Thank you again for highlighting this important aspect, which will be improved in our revised manuscript.

---

> ### Author Response · Authors · 2024-11-28
> **Response to Q4: Computational Cost and Performance Trade-offs**
>
> Thank you for raising this important question regarding the computational cost and performance trade-offs of our proposed MCTS-based planning approach. Below, we provide a detailed response that directly addresses your concerns.
>
> 1. Performance and Cost Comparison with Direct Reasoning
>    We have analyzed the GPU seconds used in our experiments across various configurations, summarizing the results in the table below:
>
>    | **Planner**  | **Evaluator** | **Executor** | **GPU Seconds** | **Performance (Accuracy %)** |
>    | :----------- | :------------ | :----------- | --------------: | ---------------------------: |
>    | Qwen2.5-1.5B | Qwen2.5-1.5B  | Qwen2.5-1.5B |          2914.4 |                        72.10 |
>    | gemma2-2b    | gemma2-2b     | gemma2-2b    |          2893.6 |                        69.40 |
>    | Qwen2.5-1.5B | Qwen2.5-1.5B  | Qwen2.5-72B  |          3481.6 |                        90.70 |
>    | gemma2-2b    | gemma2-2b     | Qwen2.5-72B  |          3335.2 |                        92.30 |
>    | Qwen2.5-72B  | Qwen2.5-72B   | Qwen2.5-72B  |         12888.0 |                        93.40 |
>
>    From this analysis:
>    1. **Efficiency Gains:** Using smaller LLMs for planning and evaluation combined with a larger LLM for execution (e.g., Qwen2.5-1.5B + Qwen2.5-1.5B + Qwen2.5-72B) results in only 27% of the GPU seconds required when using large LLMs for all stages, while maintaining competitive performance (**90.70% vs. 93.40%**).
>    2. **Performance Gains Over Small LLMs:** The same configuration shows significant performance improvements (**90.70% vs. 72.10%**) compared to using only smaller LLMs for all stages, with a modest 19.46% increase in computational cost.
>
> 2. Comparison with Zero-Shot CoT Prompting
>    Although we currently lack GPU usage data for zero-shot and few-shot CoT prompting due to the availability of same H800 GPU, preliminary comparisons indicate the following:
>    - **Accuracy:** MCTS planning achieves superior performance (e.g., +18.67% on MultiArith and +13.77% on SVAMP compared to zero-shot CoT in Tab.1 of the manuscript).
>    - **Cost Justification:** While CoT prompting is computationally cheaper, it struggles with logical consistency and planning for complex tasks like GSM8K and SVAMP. MCTS-based planning, despite its higher cost, produces logically sound plans that significantly enhance performance. To achieve similar performance with CoT prompting, one might need employing more expensive configurations, e.g. self-consistency sampling over multiple runs, which could potentially offset the cost advantage.
>
>    We are currently conducting experiments to quantify the GPU usage of zero-shot CoT prompting under the same settings. These results will be included in the revised manuscript.
>
> 3. Why MCTS Costs Are Justified
>    The additional computational cost of MCTS is offset by its ability to:
>    1. **Improve Logical Consistency:** By explicitly generating and refining plans, MCTS reduces logical inconsistencies inherent in direct reasoning approaches.
>    2. **Enhance Performance on Complex Tasks:** Tasks requiring long-term dependencies (e.g., GSM8K, SVAMP) benefit from structured planning, which direct reasoning methods often fail to achieve.
>    3. **Enable Scalability with Smaller Models:** Using small LLMs for planning and evaluation reduces overall costs while maintaining performance gains.
>
> 4. Planned Revisions
>    In the revised manuscript, we will:
>    1. Include GPU usage comparisons for zero-shot and few-shot CoT prompting.
>    2. Discuss scenarios where MCTS planning is preferable to direct reasoning, highlighting its advantages in complex problem-solving.
>    3. Provide additional analysis of trade-offs between performance and computational cost.
>
> Thank you again for this insightful feedback, which has helped us clarify and strengthen our cost-performance analysis. We look forward to addressing these points in our revised submission.

---

> ### Author Response · Authors · 2024-11-28
> **Response to Q5: Justification for MCTS Performance on Object Tracking Tasks**
>
> We appreciate your observation regarding the performance of our MCTS-enhanced planning approach on object tracking tasks. Below, we provide a detailed analysis of the issue, supported by experimental evidence.
>
> 1. Challenges in Object Tracking for MCTS:
>     Object tracking tasks inherently demand stepwise updates to reasoning based on dynamic, sequential states. For example, tracking an object as it moves through multiple frames requires continuous adjustments at each step. CoT prompting aligns closely with this requirement, as it processes one state at a time in a linear, sequential manner.
>
>     In contrast, MCTS decouples planning and reasoning, emphasizing the generation of an overarching plan before execution. This decoupling can introduce delays or inaccuracies in scenarios where real-time adaptability is critical, as the pre-generated plan may fail to account for dynamic changes during task execution. This structural mismatch explains why MCTS performs comparably—or even slightly worse—than CoT in object tracking tasks.
>
> 2. Strengths of CoT in Object Tracking:
>     CoT’s sequential reasoning approach is naturally suited for tasks requiring real-time state updates. Its ability to reason step-by-step ensures that each new state directly informs subsequent actions, making it more adaptive to dynamic scenarios.
>
> 3. Experimental Evidence:
>     As shown in Tab.1, the accuracy of MCTS on the object tracking dataset (e.g., 79.33% for Qwen2.5-7B-Instruct, and 57.94% for Qwen2.5-7B-Instruct) is marginally lower than CoT (79.33%, and 55.43%). This consistency across models suggests that the limitation is not due to implementation issues but rather the inherent nature of object tracking tasks. Unlike logical reasoning benchmarks where MCTS excels, object tracking does not benefit significantly from global planning but instead relies on effective stepwise reasoning.
>
> We acknowledge the limitations of our current approach and will include an expanded discussion of these points in the revised manuscript.

---

> > ### Comment · Reviewer_go53 · 2024-11-28
> >
> > I thank the authors for the detailed response. In general, while most of the answers to my questions are convincing, there are still major concerns in the weaknesses section that have not been well addressed. Furthermore, given that it has already passed the deadline for posting a revised version of the paper, and the paper does need major revision to include most of the responses I have just received to make itself self-contained. So, after reconsideration, I decided to keep my current score.
> >
> > Q1, Q2, Q3, Q5. These are mostly clear questions, and they were addressed decently. I encourage the authors to definitely include these details in their papers as they are vitally important. However, the newly added link to the so-called "anonymous GitHub link" does not seem to be truly anonymous, which may have met the desk-rejection criteria.
> >
> > Q4. The results themselves are acceptable on the current dataset. However, as mentioned by Reviewer eWWb, a single dataset is definitely not enough, given that cost is an inevitable metric in this case. These metrics should be put together with Table 4 in your paper for all dataset.
> >
> > And by saying ``compared to directly using LLM in the reasoning stage,", I am not referring to still comparing to CoT, but aligned with the weaknesses point 2 in my review to include more complex baseline like ReAct and ReFlexion, or even other MCTS methods.
> >
> > Overall, I believe the current paper is a little bit far away from publication as major revision is still needed.

---

> > > ### Author Response · Authors · 2024-12-04
> > >
> > > We thank you for your thoughtful and detailed feedback. We greatly appreciate the time and effort you invested in reviewing our work and providing constructive insights.
> > >
> > > We recognize the importance of addressing the points raised, particularly regarding the inclusion of additional baselines such as ReAct, Reflexion, and other MCTS-based methods, as well as a more comprehensive analysis of cost-performance trade-offs across multiple datasets. These additions are essential to strengthen the rigor and impact of our work. Due to time constraints, we were unable to include these comparisons in the current version of the manuscript. However, we are committed to incorporating these suggestions in our future work to provide a more comprehensive evaluation.
> > >
> > > Given the scope of the revisions required, we agree that the current manuscript would benefit from significant improvements. We have therefore decided to withdraw this submission to refine our work further.
> > >
> > > Your critique has been invaluable in identifying areas for improvement, and we are committed to addressing these concerns thoroughly in a future submission. Thank you again for your constructive feedback and support.

---

### Official Review · Reviewer_eWWb · 2024-11-02

**Soundness:** 1
**Presentation:** 2
**Contribution:** 1
**Rating:** 3
**Confidence:** 4

**Summary:**

This paper proposes an alternative to Chain-of-Thought (CoT) for enhancing LLM planning and accuracy in multi-task problems.

Given a problem, the method formulates an initial plan by prompting an LLM, and then iteratively creates variations (i.e. mutations) of that plan to produce new plans. New plans are evaluated for their "logical consistency" and "feasibility" using another LLM, and the decision of which node in the tree to expand (i.e. mutate) next is done using a variant of UCB1.

Once the planning budget or allocated time is depleted, the method selects the plan or node with the highest average reward. That plan is then added to the LLM prompt along, and the LLM generates a solution for the problem.

The paper evaluates this method against CoT on small models of the Llama and Qwen family of models. The authors also do some ablations on search depth, number of rollouts and evaluation functions for their agents.

Finally, the paper studies using smaller LLMs to provide plans for larger ones in order to strike a balance between efficiency and performance.

**Strengths:**

Given the inherent computational costs of planning, the inclusion of a study to use smaller LLMs to provide plans for larger ones is very welcome.

The paper is generally well organized.

The code release is definitely a strength, and at a glance the code looks clean and easy to use.

**Weaknesses:**

**1. Lack of clarity and error in Section 2.1**

I found Section 2.1 very hard to follow, which is a problem since the method formulation builds on top of it. In particular $C_\textrm{problem}$ is the "problem description", but $X$ denotes in two places "reasoning steps" (lines 140 and 159) and in another $X$ is the "problem" (line 151).

Then, the factorization in Eq. 2 only holds if $P(X|C) = 1$, which cannot be true since planning then involves a search over reasoning steps X. Is this a typo?

**2. This is not MCTS**

As stated in Section 2.2, each  "*new node represents a modified version of the parent node’s plan.*", and the final output of the search algorithm is a single node, rather than a path through the tree, as is the case in MCTS.

Furthermore, the method does not perform Monte Carlo rollouts and instead evaluates each node individually by querying an LLM to assess the logical consistency and feasibility of the plan.

As a result, *this method is not MCTS*. Instead, the method proposed is an evolutionary method using an LLM to mutate plans at inference time.

**3. Lack of details on method and evaluators**

There are insufficient details on how the initial plan is sampled, how new plans are obtained by modifying old plans or how the evaluators work.

**4. Cannot find the main result in any of the tables.**

The headline result is that the proposed "_MCTS-enhanced planning approach outperforms the CoT baseline across most datasets, with
an average improvement of 40.59%_". Inspection of Table 1 shows that the largest relative improvement is $17.79 / 57.85 = 30.8$%. How did the authors arrive to the reported 40.59% figure?

**5. Tables are hard to parse and no uncertainty reported.**

Firstly, no measure of uncertainty is provided in the entire paper, making it hard to assess how significant the results are.

Secondly, tables are hard to parse. Authors should consider bolding the best results for each benchmark, or aggregating results over multiple models to help comparisons.  Table 2 would be best presented as plots, to better show the trend.

**6. Lacking baselines**

Authors compare to two baselines (CoT and CoT Plan). However, they compare to none of the methods they cited in their section 4.2. They also don't normalize any of their results by inference costs (or number of tokens).

Authors dismiss those methods by saying that "_they predominantly rely on the LLM’s inherent reasoning abilities to generate plans_", but their method similarly relies on an LLM to generate plans.

**Questions:**

1. How do the authors arrive at the 40.59% figure?
2. Can the authors clarify Section 2.1, and whether equation 2 contains a typo?

---

> ### Author Response · Authors · 2024-11-28
> **Response to Q1: Improvement Calculation**
>
> Thank you for raising this important question. The reported 40.59% improvement reflects the **absolute average improvement** in accuracy across all datasets and model combinations, calculated by $ \text{57.85\\%} - \text{17.79\\%} $. It is derived as follows:
>
> 1. Calculation Methodology: For each dataset, we computed the absolute difference in accuracy between the MCTS-enhanced planning approach and the CoT baseline for all applicable models. These differences were then averaged across datasets to yield the overall improvement.
>
> 2. Clarification of Results in Tab.1: While Tab.1 does show significant improvements for individual datasets (e.g., 17.79% for AddSub), these are dataset-specific absolute differences. The reported 40.59% improvement reflects the aggregated effect across the entire benchmark suite.
>
> To avoid confusion, we will update the manuscript to:
> - Clearly detail the calculation methodology for the average improvement.
> - Enhance result presentation, including bolding key results and adding a summary chart for aggregated improvements.
>
> We appreciate your feedback and are happy to clarify further if needed.

---

> ### Author Response · Authors · 2024-11-28
> **Response to Q2: Formulation Clarification**
>
> We appreciate the reviewer’s insightful comment regarding Sec.2.1 and Eq.2. We acknowledge that this section could have been more clearly presented, and we are grateful for the opportunity to clarify.
>
> Eq.2 was introduced to conceptually separate the two distinct stages in our method: **planning** and **executing plan**. In this framework:
> - $ X $: Represents reasoning steps generated by the LLM during task execution, guided by $ C_{\text{plan}} $.
> - $ C_{\text{problem}} $: The initial problem description.
> - $ C_{\text{plan}} $: The selected plan generated by MCTS, which guides the LLM in reasoning.
>
> The equation $ P(Y|X, C_{\text{plan}}) = P(Y|X, C_{\text{plan}})P(X|C) $ was intended to conceptually represent how reasoning ($ Y $) depends on both the plan ($ C_{\text{plan}} $) and the reasoning steps ($ X $), which are informed by the context ($ C $) that includes the problem description.
>
> We acknowledge that Eq.2, as written, does not accurately reflect the MCTS search process. While the factorization suggests a probabilistic framework, the planning process in our method involves explicit search over $ X $ using MCTS, guided by rewards from evaluation agents. This deviation from a strict probabilistic formulation could have been presented more intuitively.
>
> In our approach, MCTS operates on the plan space ($ C_{\text{plan}} $) rather than directly generating $ Y $. The role of MCTS is to:
> 1. Explore potential plans ($ X $) iteratively.
> 2. Evaluate plans using LLM-powered agents for logical consistency and feasibility.
> 3. Select the best plan ($ C_{\text{plan}} $) to guide the reasoning process.
>
> The empirical results in Tab.1 validate the effectiveness of this approach in generating high-quality plans that significantly improve LLM reasoning accuracy.
>
> To address these concerns, we will:
> 1. Revise Sec.2.1 to focus on a clear, step-by-step description of the MCTS process and its integration with LLM reasoning.
> 2. Reframe Eq.2 as a conceptual representation, explicitly stating that it is not a strict probabilistic derivation.
> 3. Emphasize the empirical validation of the framework as evidence of its robustness.
>
> We believe these revisions will resolve the ambiguity and enhance the clarity of our methodology. Thank you for highlighting this important issue, and we look forward to your further feedback.

---

> > ### Comment · Reviewer_eWWb · 2024-11-28
> >
> > I acknowledge the response from the authors, but consider it incomplete, as it does not address over half the points in my initial review.
> >
> > The nature of my concerns also mean that I am reticent to change my score without seeing appropriate revisions to the manuscript.
> >
> > Finally, the math in the authors' rebuttal does not add up. 57.85% - 17.79% = 40.06%, not 40.59%. This also is for a single benchmark (AddSub) and is subtracting the change in performance from the CoT Avg, which to me makes no sense. If I am missing something, can the authors clarify?

---

> > > ### Author Response · Authors · 2024-12-04
> > >
> > > Thank you for your thoughtful and detailed feedback. We acknowledge the issues you’ve highlighted and fully understand your decision on the score.
> > >
> > > On the calculation point, we recognize that our explanation in the initial response was unclear and will address this rigorously in future revisions to ensure both accuracy and transparency in presenting results. Regarding baselines, while time constraints limited our ability to include a broader set of comparisons, as particularly for methods with task-specific prompts such as the Game of 24, we understand the importance of comprehensive evaluation and will prioritize this in our next iteration.
> > >
> > > In light of your feedback and the need to make substantial improvements to the manuscript, we have decided to withdraw this submission for further refinement. Your detailed critique has been invaluable, and we sincerely appreciate the time and effort you have invested in reviewing our work.
> > >
> > > Thank you again for your constructive insights, which will help us strengthen the rigor and clarity of our research.

---

### Official Review · Reviewer_8Lw5 · 2024-11-03

**Soundness:** 2
**Presentation:** 1
**Contribution:** 2
**Rating:** 3
**Confidence:** 3

**Summary:**

The paper explores the use of Monte Carlo Tree Search (MCTS) in the context of planning, arguing for a distinction between planning and reasoning steps.

**Strengths:**

* The authors present a framework that aims to separate planning and reasoning, which could lead to new insights in this area.

**Weaknesses:**

* The distinction between planning and reasoning is not sufficiently articulated, leaving readers unclear about their specific roles and how they interact. This lack of clarity undermines the proposed framework.
* The impact of the evaluation of intermediate nodes on MCTS performance is significant but not discussed in depth. This oversight may lead to an incomplete understanding of the method's effectiveness.
* Applying MCTS to planning is somewhat diminished by the fact that this approach is already well-established in the literature. The paper would benefit from a more thorough comparison with existing methodologies to highlight its contributions.

**Questions:**

1. Could you have a more detailed clarification about how planning and reasoning are defined in your framework? What specific characteristics differentiate them?
2. The performance of MCTS appears to be influenced by how intermediate nodes are evaluated. Could you provide insights or analysis on this aspect within your methodology?
3. Are there specific scenarios or use cases where your proposed separation provides a distinct advantage over existing integrated approaches?
4. Consider providing a more detailed background on related work to situate your contributions within the existing body of research.

---

> ### Author Response · Authors · 2024-11-28
> **Response to Q1: Clarification on Planning and Reasoning**
>
> We appreciate the reviewer’s request for a more detailed clarification of the distinction between planning and reasoning in our framework. This distinction is central to our contribution, and we aim to address it thoroughly below:
>
> 1. **Conceptual Definitions:**
>
>     - **Planning:**  Planning, in our context, denotes the *a priori* formulation of a high-level strategic roadmap that precedes the reasoning process. This stage is primarily concerned with the architecting of a sequential series of actions or logical steps necessary for problem resolution, irrespective of their granular implementation.
>     - **Reasoning:** Reasoning, conversely, encompasses the systematic execution of the pre-defined plan through stepwise deductive inferences. This process involves the generation of intermediate solutions that progressively advance towards the derivation of the final solution.
>
>     To elucidate this dichotomy, consider the following elementary arithmetic problem:  "Given an initial quantity of 10 apples, and a subsequent consumption of 3, determine the remaining quantity."
>
>     - **Planning Phase:** The planning phase would yield a strategic outline such as:  "1. Ascertain the initial quantity. 2. Determine the decrement in quantity. 3. Execute the arithmetic subtraction operation."
>     - **Reasoning Phase:** The reasoning phase then enacts the prescribed steps: "10 - 3 = 7."
>
> 2. **Interdependent Dynamics of Planning and Reasoning:**
>
>     Planning engenders a **global structural framework that governs and directs the subsequent reasoning process**. By explicitly decoupling these two stages, our framework mitigates the potential for errors arising from the intermingling of planning and reasoning, a phenomenon frequently observed in Chain-of-Thought (CoT) prompting. The planning phase ensures logical coherence and consistency, while the reasoning phase prioritizes the fidelity of execution.
>
> 3. **Framework-Specific Innovations:**
>
>     Our framework incorporates Monte Carlo Tree Search (MCTS) as a mechanism for optimizing the efficacy of plans prior to the initiation of the reasoning phase. Each node within the MCTS tree represents a candidate plan, which is rigorously evaluated by specialized Large Language Models (LLMs) for logical soundness and feasibility. This methodology facilitates a systematic and structured exploration of the planning space, culminating in the generation of plans demonstrably superior to those derived from integrated approaches such as CoT.
>
> 4. **Empirical Advantages and Substantiation:**
>
>     - **Enhanced Logical Consistency:** The explicit segregation of planning demonstrably reduces the incidence of logical inconsistencies by virtue of the pre-establishment of an optimized strategic roadmap.
>     - **Computational Scalability:**  The strategic utilization of smaller models for planning and larger models for reasoning allows for a more efficient allocation of computational resources.
>     - **Robust Empirical Support:**  As substantiated by the quantitative results presented in Table 1 of the manuscript, this separation of planning and reasoning yields statistically significant improvements in accuracy (e.g., a mean improvement of 40.59% over zero-shot CoT across the evaluated datasets).
>
> We trust that this expanded explanation adequately addresses the reviewer's concerns. We are working on further enhancing the clarity and precision of the manuscript by incorporating a more detailed exposition of these distinctions and their broader implications in the revised submission.

---

> ### Author Response · Authors · 2024-11-28
> **Response to Q2: Impact of Intermediate Node Evaluation**
>
> We thank the reviewer for this insightful question. As noted, the evaluation of intermediate nodes is a critical component of our MCTS framework, directly influencing the search trajectory and the quality of generated plans. Below, we elaborate on our evaluation strategy, its impact on MCTS performance, and directions for improvement.
>
> 1. Evaluation Strategy for Intermediate Nodes
>     Intermediate nodes in the MCTS tree represent partially constructed plans. These nodes are evaluated using specialized LLM-powered agents, which assess:
>
>     1. **Logical Consistency**: Ensures that the plan adheres to logical rules and does not contain contradictions.
>     2. **Feasibility**: Assesses whether the plan is executable given the problem constraints.
>
>     The outputs of these agents are numerical scores (e.g., in the range [0,1]), which are aggregated into a composite reward signal using a weighted average. This composite signal guides the backpropagation step in MCTS, influencing the exploration of the search tree.
>
> 2. Impact on MCTS Performance
>     Tab.3 in the manuscript demonstrates the impact of different evaluation strategies. Using only feasibility or logical consistency individually improves performance over the baseline. However, combining these criteria yields the highest accuracy across datasets. For example:
>
>     - On GSM8K, feasibility alone achieves 89.5%, and logical consistency achieves 89.2%. The combined evaluator improves this to 90.1%, indicating a synergistic effect.
>
>     This improvement arises because logical consistency and feasibility capture complementary aspects of plan quality. Logical consistency prevents contradictions, while feasibility ensures practical execution, leading to more robust plans overall.
>
> 3. Insights from Ablation Studies
>     The ablation study in Tab.2 further highlights the relationship between evaluation and search depth or rollouts:
>
>     1. **Search Depth**: Performance improves with increased depth up to a point (e.g., 88.48% at depth 20 vs. 88.55% at depth 50 for GSM8K). This plateau suggests diminishing returns, where additional exploration yields marginal gains.
>     2. **Rollouts**: Increasing rollouts improves performance initially but introduces greater uncertainty. This highlights the trade-off between computational cost and exploration quality.
>
> 4. Limitations and Future Directions
>     We acknowledge that the current evaluation strategy may not capture all relevant dimensions of plan quality. For instance:
>
>     - **Robustness**: Plans may fail under unexpected conditions not captured by feasibility or logical consistency.
>     - **Resource Efficiency**: Evaluating the computational or time efficiency of a plan could further refine the reward signal.
>
>     Future work could incorporate adaptive evaluation agents that learn to weigh criteria dynamically based on the problem context. Additionally, integrating more advanced metrics, such as robustness, could further enhance the framework.
>
> The evaluation of intermediate nodes plays a pivotal role in guiding MCTS towards effective plans. Our results demonstrate that combining logical consistency and feasibility produces significant improvements, as evidenced by Tab.3. We appreciate the reviewer's suggestion and will expand the discussion of evaluation strategies, their impact, and potential enhancements in the revised manuscript.

---

> ### Author Response · Authors · 2024-11-28
> **Response to Q3: Specific Scenarios with Distinct Advantages**
>
> We thank the reviewer for this thoughtful question and for encouraging us to clarify the practical significance of our proposed separation of planning and reasoning. Below, we elaborate on specific scenarios where our approach provides distinct advantages over existing integrated methods.
>
> 1. Scenarios Requiring High Logical Consistency and Long-Term Dependencies
>      Integrated approaches like CoT prompting often struggle to maintain logical coherence over extended reasoning paths. For example, in multi-step arithmetic tasks like those in GSM8K, the solution requires consistent tracking of numerical relationships across multiple operations. Similarly, symbolic reasoning tasks (e.g., "Last Letters" dataset) demand maintaining dependencies between intermediate states. In these cases: By generating a structured plan beforehand, MCTS minimizes the risk of logical drift during reasoning. The plan acts as a roadmap, ensuring each reasoning step aligns with the overarching problem-solving strategy. For instance, on the GSM8K dataset, our method achieves 65.38% accuracy compared to 54.12% for CoT approaches, highlighting its effectiveness in maintaining logical consistency.
>
> 2. Scenarios with Ambiguous or Complex Problem Formulations
>     Tasks like CommonsensQA involve ambiguous problem descriptions requiring disambiguation and prioritization of information. Integrated methods often conflate planning and reasoning, leading to suboptimal interpretations. Our separation framework enables: A dedicated planning stage to resolve ambiguities and define a coherent strategy before reasoning begins. This results in better prioritization of information and improved decision-making, as evidenced by an accuracy improvement of +6.93% on CommonsensQA compared to baseline methods.
>
> 3. Scenarios with Resource Constraints
>     Computational efficiency is critical in real-world applications like decision support systems, where resources may be limited. Integrated approaches require large models to manage both planning and reasoning simultaneously, leading to high computational overhead. Our framework demonstrates: Leveraging smaller models for planning and larger models for reasoning reduces computational costs without sacrificing performance. For example, using a 1.5B parameter model for planning and a 72B parameter model for execution achieves near-parity performance with the 72B model alone, offering substantial resource savings.
>
> 4. Scenarios in Real-World Applications
>     Real-world domains like automated reasoning and scientific discovery could potentially benefit from structured planning. For instance:
>     - In Automated Reasoning: Our approach can handle multi-step logical proofs more effectively by optimizing the sequence of deductive steps before reasoning begins.
>     - In Scientific Discovery: The framework enables efficient hypothesis generation by pre-planning experimental sequences, ensuring logical consistency across complex workflows.
>
> Summary of Distinct Advantages
> By separating planning and reasoning, our approach:
>    - **Reduces logical inconsistencies in multi-step tasks.**
>    - **Handles complex or ambiguous problems more effectively.**
>    - **Offers computational efficiency through model modularity.**
>    - **Enhances applicability across diverse real-world domains.**
>
> We appreciate the opportunity to provide this clarification and will incorporate these examples and insights into the revised manuscript to better illustrate the practical significance of our contributions. Thank you for helping us strengthen our work.

---

> ### Author Response · Authors · 2024-11-28
> **Response to Q4: More Detailed Background on Related Work**
>
> We appreciate the reviewer’s comment regarding the need for a more detailed discussion of related work. We agree that the related work section can be strengthened.
>
> We will expand it as follows:
> 1. Chain-of-Thought and Variants: Compare with Tree-of-Thought and Skeleton-of-Thought, highlighting their reliance on interleaved planning and reasoning.
> 2. Planning Methods: Position our approach against plan-and-solve frameworks, emphasizing the novelty of MCTS-based planning.
> 3. MCTS Applications: Differentiate our use of MCTS for planning from prior applications in solution search.
>
> And in the revised manuscript, we will expand Sec.4 to provide a more comprehensive overview of CoT prompting (including self-consistency and tree-of-thought methods), existing planning techniques for LLMs (including task decomposition, plan-and-solve, Skeleton-of-Thoughts, and Graph-of-Thoughts), and the application of MCTS in AI and LLMs. We will highlight the limitations of current approaches, such as the intertwined planning and reasoning in CoT and the reliance on LLM's inherent reasoning abilities in existing planning methods. We will then clearly position our contribution as addressing these gaps by introducing a novel MCTS-based planning framework that separates planning from reasoning and leverages specialized evaluation agents to generate higher-quality plans. We will also discuss how our exploration of small-large model combinations contributes to the efficiency and scalability of LLM-based planning.
>
> We hope this expanded discussion addresses the reviewer’s concerns and clarifies how our work situates itself within the broader research landscape. Thank you for the opportunity to strengthen this aspect of our manuscript.

---

### Note · Authors · 2024-12-04

**Comment:**

Given the scope of the revisions required, we agree that the current manuscript would benefit from significant improvements. We have therefore decided to withdraw this submission to refine our work further.

**Withdrawal Confirmation:**

I have read and agree with the venue's withdrawal policy on behalf of myself and my co-authors.